# In Silico Characterization of Calcineurin from Pathogenic Obligate Intracellular Trypanosomatids: Potential New Biological Roles

**DOI:** 10.3390/biom11091322

**Published:** 2021-09-07

**Authors:** Patricio R. Orrego, Mayela Serrano-Rodríguez, Mauro Cortez, Jorge E. Araya

**Affiliations:** 1Departamento Biomédico, Facultad de Ciencias de la Salud, Universidad de Antofagasta, Antofagasta 1270300, Chile; 2Departamento de Tecnología Médica, Facultad de Ciencias de la Salud, Universidad de Antofagasta, Antofagasta 1270300, Chile; mayela.serrano@uantof.cl; 3Departamento de Parasitologia, Instituto de Ciências Biomédicas, Universidade de São Paulo, São Paulo 05508-000, Brazil; mcortez@usp.br; 4Center for Biotechnology and Bioengineering, CeBIB, Universidad de Antofagasta, Antofagasta 1270300, Chile

**Keywords:** calcineurin, in silico analysis, intracellular trypanosomatids, *Leishmania*, *Trypanosoma cruzi*

## Abstract

Calcineurin (CaN) is present in all eukaryotic cells, including intracellular trypanosomatid parasites such as *Trypanosoma cruzi* (*Tc*) and *Leishmania* spp. (*L*spp). In this study, we performed an in silico analysis of the CaN subunits, comparing them with the human (*Hs*) and looking their structure, post-translational mechanisms, subcellular distribution, interactors, and secretion potential. The differences in the structure of the domains suggest the existence of regulatory mechanisms and differential activity between these protozoa. Regulatory subunits are partially conserved, showing differences in their Ca^2+^-binding domains and myristoylation potential compared with human CaN. The subcellular distribution reveals that the catalytic subunits *Tc*CaNA1, *Tc*CaNA2, *L*sppCaNA1, *L*sppCaNA1_var, and *L*sppCaNA2 associate preferentially with the plasma membrane compared with the cytoplasmic location of *Hs*CaNAα. For regulatory subunits, *Hs*CaNB-1 and *L*sppCaNB associate preferentially with the nucleus and cytoplasm, and *Tc*CaNB with chloroplast and cytoplasm. Calpain cleavage sites on CaNA suggest differential processing. CaNA and CaNB of these trypanosomatids have the potential to be secreted and could play a role in remote communication. Therefore, this background can be used to develop new drugs for protozoan pathogens that cause neglected disease.

## 1. Introduction

People all over the world are affected by leishmaniasis and American trypanosomiasis, two neglected tropical diseases that infect over 6 and 12 million people, respectively [1,2,3]. Several issues, including a lack of safe/optional drugs, parasite resistance, and ineffective insect vector control, have caused researchers to race to develop more effective treatments for diseases [4], increasing the current alternative studies with promising results [5,6,7]. These vector-borne diseases are caused by protozoan parasites of the Trypanosomatidae family, *Leishmania* spp. (*L*spp) and *Trypanosoma cruzi* (*Tc*), which have unique characteristics. Insect vectors release infective parasite forms during the blood meal in humans, where they face harsh extracellular conditions and quickly invade cells to avoid extracellular immune response [8,9]. It is well established that these intracellular parasites infect different cells, activating the release of Ca^2+^ from intracellular stores [10,11] from both parasite and host cells, stimulating different signaling pathways that promote critical interactions. Ca^2+^-dependent phosphatases have thus emerged as critical regulator molecules [12] for intracellular trypanosomatids.

Calcineurin (CaN, also called protein 2B phosphatase, PP2B) is the main binding enzyme to the multifunctional protein calmodulin (CaM) in the brain and the only serine/threonine phosphatase under the control of Ca^2+^/CaM, playing a critical role in Ca^2+^-mediated cellular responses [13,14,15,16]. Stimulation of CaN by CaM ensures the coordinated regulation of its phosphatase function with the activities of many other enzymes, including Ca^2+^ and CaM control-dependent kinases. CaN is widely distributed, despite its abundance in nervous tissues, and its structure is highly conserved from yeast to human [15]. CaN is a heterodimer composed of two subunits: a catalytic subunit A (CaNA) and a regulatory subunit B (CaNB) [17].

CaN biology is distinguished by the specificity of its substrates, which include transcription factors, specific inhibitors, ion channels, apoptotic molecules, and cytoskeletal proteins [18]. In mammals, the CaNA active site has a mass ranging from 57 to 59 kDa depending on the isoform [17]. Its amino acid structure is highly conserved in eukaryotes, as all CaNA genes encode a polypeptide consisting of a catalytic domain, homologous to other protein phosphatases, and three regulatory domains at the C-terminus that distinguish CaN from other members of the PPP family such as PP1 and PP2C [17]. These three domains have been identified as the CaNB-binding domain [19,20], the CaM-binding domain [21,22], and the autoinhibitory domain (AID) [20,23], which binds to the CaN active site in the absence of Ca^2+^/CaM, inhibiting the enzyme and acting in concert with the CaM-binding domain, regulating this last domain [24].

CaN has gained importance since its discovery as the target of the immunosuppressive drugs cyclosporin A (CsA) and tacrolimus (FK506) [25], which inhibit CaN when complexed with their respective cytoplasmic receptors (CyP) or FK506-binding proteins (FKBP), respectively, both complexes that block access to the active site of substrates [25,26,27].

The Ca^2+^-CaN signaling pathway is conserved in many eukaryotic microorganisms [28,29], making it an appealing target for the development of new anti-pathogen drugs [12]. In fact, studies in yeasts such as *Saccharomyces cerevisiae* show that CaN plays an important role in the microorganism’s survival by dephosphorylating the transcription factor Crz1, which moves it from the cytosol to the nucleus, subsequent to the activation of CaN [30]. In the case of protists, the lack of transcription factors [31] suggests that this CaN pathway operates by a different mechanism [12] in cellular processes such as adherence to the host cell, cell invasion, thermo-tolerance, and flagellar motility [12]. Notably, in human protozoan pathogens such as *T. cruzi*, the role of CaN has been linked to processes of cellular invasion of infective forms and proliferation of non-infective forms of the parasite [32,33]. CaN of *Leishmania* spp., on the other hand, has been linked to an important role in thermo-tolerance and survival processes in mammalian hosts [34], being associated with flagellar motility, response to environmental changes, survival, and virulence [35].

In this paper, we perform an in silico characterization of the catalytic and regulatory CaN subunits of the obligate intracellular trypanosomatids, *T. cruzi*, and *Leishmania* spp., comparing them to human CaN subunits.

## 2. Materials and Methods

### 2.1. Database Inspection

The sequences corresponding to the catalytic A1, A2, and CaN regulatory subunits in *T. cruzi* and *Leishmania* spp. were obtained from the NCBI protein databases (https://www.ncbi.nlm.nih.gov/ (accessed on 4 April 2020)) and TriTrypDB (https://tritrypdb.org/tritrypdb/ (accessed on 4 April 2020)) [36] and used as references; those characterized in *T. cruzi* included CAI48024 and CAI48025 [37], ABY61052 and ABO14295 [32], and ADN03392 [33] and those characterized in *Leishmania* included LmjF.26.2530 [38], LmjF.36.1980 [38,39], and LmjF.21.1630 [36].

In the case of the A1 catalytic and CaN regulatory subunits in *Homo sapiens*, the sequences NP_000935 and NP_000936 (National Center for Biotechnology Information (NCBI) were used. Bethesda (MD): National Library of Medicine (US), National Center for Biotechnology Information; (1988)—[cited 6 January 2021]. Available from https://www.ncbi.nlm.nih.gov/ (accessed on 01 January 2021)).

### 2.2. Obtaining Consensus Sequences and Multiple Alignment of CaN

The consensus amino acid sequences of the catalytic A1, A2, and CaN regulatory subunits of *T. cruzi* and *Leishmania* spp. were obtained using the EMBOSS Cons tool (https://www.ebi.ac.uk/Tools/msa/emboss_cons/ (accessed on 30 May 2020)) (Appendix A) [40]. The multiple alignment of trypanosomatids and *H. sapiens* was performed in the MEGA7 software [41] using the default ClustalW parameters. The alignments obtained were viewed in the SnapGene Viewer software (version 5.1.3, from GSL Biotech; available at snapgene.com).

### 2.3. Evaluation of Conserved Domains, Determination of Physicochemical Parameters, and Hydrophobicity Profiles

The identification of the conserved domains of the sequences under study was carried out through the Conserved Domains Database (CDD) [42,43,44] and ExPASy ScanProsite (https://prosite.expasy.org/scanprosite/ (accessed on 30 May 2020)) [45]. The domain structure was represented by PROSITE ‘MyDomains’ image creator tool [46].

The determination of the physicochemical parameters was carried out in the ProtParam tool (https://web.expasy.org/protparam/protparam-doc.html (accessed on 30 May 2020)) [47] and the hydrophobicity profiles with the Kyte–Doolittle algorithm [48], with a window of nine residues in ProtScale (https://web.expasy.org/protscale/ (accessed on 30 May 2020)) [47].

### 2.4. In Silico Identification of Protein–Protein Interactions of CaN Regulatory Subunits

The identification of interactions of the CaNB subunits of *T. cruzi* (accession number ABO14295) and *L. major* (accession number LmjF.21.1630) was carried out in the search tool for the recovery of interacting genes/proteins, STRING V.11.0 (https://string-db.org/ (accessed on 11 June 2020)) [49]. The resources used were experimental results and databases, with a value of 0.7 (high confidence) established as the minimum interaction score [49].

Similarly, the sequences of the CaN regulatory subunits of *T. cruzi* (accession number TcCLB.510519.60) and *L. major* (accession number LmjF.21.1630) were entered as proteins consulted in the TrypsNetDB databases (trypsNetDB.org) [50].

### 2.5. Prediction of the Subcellular Localization of CaN in Pathogenic Intracellular Trypanosomatids

The subcellular location of the CaN subunits in *T. cruzi* and *Leishmania* spp. was carried out in WoLF PSORT (https://wolfpsort.hgc.jp/ (accessed on 11 June 2020)) [51] with Fungi selected as the type of organism; in addition, the CELLO2GO tool was also used (http://cello.life.nctu.edu.tw/cello2go (accessed on 11 June 2020)) by selecting Eukaryotes [52].

### 2.6. Prediction of Entry to the Non-Classical Secretory Pathway of CaN in Pathogenic Intracellular Trypanosomatids

The SecretomeP 2.0 Server tool (http://www.cbs.dtu.dk/services/SecretomeP/ (accessed on 11 June 2020)) was used to predict the entry of the different CaN subunits into the non-classical secretory pathway using the prediction for mammalian sequences [53] with a threshold of 0.6.

### 2.7. Prediction of Cleavage by Calpains in CaN Catalytic Subunits of Pathogenic Intracellular Trypanosomatids

The prediction of potential calpain cleavage sites was determined using GPS-CDD (version 1.0 available at http://ccd.biocuckoo.org/ (accessed on 7 July 2020)) with a high threshold with a cut-off of 0.654 [54]. As a reference sequence, the catalytic subunit alpha isoform from *H. sapiens* (accession number NP_000935) was used.

### 2.8. Prediction of Phosphorylation Sites in the CaM-Binding Domain (CaM-BD) of CaN Catalytic Subunits from Leishmania Spp.

Predicted phosphorylation sites were evaluated using NetPhos 3.1, a web server (http://www.cbs.dtu.dk/services/NetPhos/ (accessed on 27 July 2020)) [55,56]. Sites with a threshold >0.5 were included in the results.

## 3. Results

### 3.1. The CaN Subunits of Obligate Intracellular Trypanosomatids Possess Diverse, Partially Conserved Domain Architecture and Potential Calpain Cleavage Sites

The molecular masses estimated by ProtParam (https://web.expasy.org/protparam/ (accessed on 30 May 2020)) for the catalytic CaN subunits of T. cruzi are 44 kDa for TcCaNA1 and 45 kDa for TcCaNA2, while for Leishmania spp., the consensus sequences are 60 kDa for LsppCaNA1, 56 kDa for LsppCaNA1_var, and 45 kDa for LsppCaNA2 (Table 1). This reflects a diverse structural degree, evidenced in the domain structure of the deduced amino acid sequences under study (Figure 1), when compared with the catalytic sequence of human CaN with its three domains (binding to CaNBm, binding to CaM, and to AID). The TcCaNA1 and TcCaNA2 subunits possess the CaNB-binding (CaNB-BD) and catalytic domain, in the same way as LsppCaNA2 from Leishmania spp. However, LsppCaNA1 and LsppCaNA1_var possess all the domains present in human CaN. Interestingly, CaNA2 are those that have a higher pI (isoelectric point) than their A1 counterparts (4.83–5.61), with pI of 8.13 for TcCaNA2 and 6.37 for LsppCaNA2.

The potential calpain cleaved sites determined in GPS-CDD (from more sites to fewer sites) are as follows; 29 for *L*sppCaNA1, 12 for *Tc*CaNA2, 9 for LsppCaNA1_var, 5 for *L*sppCaNA2, and 3 for *Tc*CaNA1. *Hs*CaNAα has 19 sites (Figure 1).

In relation specifically to the CaNB-BD domain, the hydrophobicity profile in *Leishmania* spp. is more similar to the human subunit (*Hs*CaNAα) than in the catalytic subunits of *T. cruzi*, presenting the highest hydrophobic profile in *L*sppCaNA2 (Figure 2F), when compared with the other catalytic subunits of CaN (Figure 2). When CaNB-BD is analyzed among all sequences, we observe that there are two mainly hydrophobic regions that go from amino acid M347 to amino acid S373 (Figure 3, highlighted in squares). These hydrophobic regions are partially conserved, the first one (region 1, on the left) with a higher degree of conservation in relation to *Hs*CaNAα. Particularly, in the sequence MDVFTWSLPFV (region 1) of CaNB-BD, the amino acid V^349^ changes in every catalytic subunit of intracellular trypanosomatids, with only one residue (L^369^) being conserved in the second hydrophobic region (region 2, EMLVNVLNICS) of *Hs*CaNAα, except in *Tc*CaNA1 (Figure 3).

In the case of the CaM-BD domain in *Hs*CaNAα (ITSFEEAKGLDRINERMPPRRDA), it is only present in *Leishmania* in the *L*sppCaNA1 and *L*sppCaNA1_var subunits with a level of identity ranging from 42–45%. Regarding the autoinhibitory domain (AID), the level of identity is considerably lower, being 17–22% in the *L*sppCaNA1 and *L*sppCaNA1_var subunits (Figure 4), lacking the important residues for the autoinhibitory function, D488 and A489 (dotted line box). The analyzed domain structure suggests that, in the case of *Leishmania* CaNA1, they would have the complete domain structure; however, the role of AID in these subunits is unknown.

### 3.2. The CaM-Binding Domain in Leishmania Spp. Has Greater Potential to Be Regulated by Phosphorylation Than Its Human Ortholog

In total, two serines (position 21 and 27) have the potential to be phosphorylated in the CaM-BD of *Hs*CaNAα (Figure 5A), different from that observed in the domain present in *Leishmania* (*L*sppCaNA1 and *L*sppCaNA1_var), presenting a serine and a threonine at positions 17 and 22, respectively (Figure 5B). These sites can be potentially phosphorylated by protein kinase A (PKA) or by an unspecified kinase (unsp) in *Hs*CaNAα and *L*sppCaNA1 (or also for *L*sppCaNA1_var), also including in the latter the potential action of PKC or casein kinase II (Table 2).

### 3.3. CaN Regulatory Subunits of Obligate Intracellular Trypanosomatids Differ in Their Calcium-Binding Domains from Their Human Counterpart and Myristoylation Potential, but Preserve Some Canonical EF-Loops and the Docking Site for Immunophilin–Immunosuppressive Drug Complexes

As has been described [57], *Hs*CaNB-1 and *L*sppCaNB (based on the sequences obtained *L. major*) have four EF hand motifs, while *Tc*CaNB has only three. *Hs*CaNB-1 and *L*sppCaNB have two low loops and two high affinity loops for Ca^2+^, while *Tc*CaNB only has one of each type (Figure 6 and Appendix A). The complete loops, based on the *Hs*CaNB-1 sequence, meet the condition of having the amino acid organization distributed as follows: DX-[DNS]-{ILVFYW}-[DENSTG]-[DNQGHRK]-{GP}-[LIVMC]-[DENQSTAGC]-x (2)-[DE]. Moreover, the *Tc*CaNB and *L*sppCaNB subunits have a hydrophobic groove for the union of the immunophilin–immunosuppressive drug complexes, highlighting the conservation of this region that goes from M^119^ to L^124^ (hydrophobic groove), corresponding to *Hs*CaNB-1, being much more hydrophobic in *Tc*CaNB and *L*sppCaNB than in *Hs*CaNB-1, determined by the grand average of hydropathicity index (GRAVY), which is used to represent the hydrophobicity value of a peptide; positive GRAVY values indicate hydrophobic and negative values mean hydrophilic (Figure 7).

Regarding the potential myristoylation in the intracellular trypanosomatid CaNB subunits, only *Tc*CaNB has a glycine (G) in the second position in the N-term in the same way as the *Hs*CaNB-1 subunit. Curiously, despite the fact that the domain structures between *Hs*CaNB-1 and *L*sppCaNB are similar to each other, CaNB in *Leishmania* spp. lacks G in its N-terminal region to be myristoylated (Figure 6 and Figure 7, Appendix A).

When a Myristoylator (trained to predict myristoylation in the N-terminal end of the amino acid chain) was used to analyze possible myristoylation, only *Hs*CaNB-1 was predicted to be potentially myristoylated with a score of 0.98984294 (high confidence), while *Tc*CaNB was predicted to be non-myristoylated with a score of −0.146144 (Appendix A).

### 3.4. CaN Regulatory and Catalytic Subunits of Obligate Intracellular and Human Trypanosomatids Would Share the Subcellular Distribution Pattern, with Some Exceptions

The analysis of all the sequences of the catalytic subunits (CaNA) under study using WoLF SPORT suggests a distribution in the cytosolic (cyto), nuclear (nucl), cytosol, and nucleus (cyto_nucl) compartments, and exclusively with a predisposition also at the mitochondrial (mito) level in only the catalytic subunits of *T. cruzi* and *Leishmania* spp. (Table 3A), while the *Hs*CaNB-1 subunit has a distribution mainly in the cytosolic (cyto) and cyto_nucl compartments, contrasting with *Tc*CaNB and *L*sppCaNB having a marked distribution at the nuclear level (nucl) (Table 3B).

When the analyses were carried out in the CELLO2GO program for the catalytic subunits, the following results were obtained (Figure 8A): *Hs*CaNAα with 52% cytoplasmic and 5% nuclear; *Tc*CaNA1 with 70% plasmamembrane, 14% cytoplasmic, and 2% nuclear; *Tc*CaNA2 with 27% plasmamembrane, 17% extracellular and cytoplasmic, and 12% nuclear; *L*sppCaNA1 with 51% plasmamembrane, 21% cytoplasmic, and 10% nuclear; *L*sppCaNA1_var with 48% plasmamembrane, 18% cytoplasmic, and 9% nuclear; and LsppCaNA2 with 57% plasmamembrane, 25% extracellular, 7% cytoplasmic, and 3% nuclear. In the case of regulatory subunits (Figure 8B), localization of *Hs*CaNB-1 with 39% cytoplasmic and 54% nuclear; *Tc*CaNB with 15% plasmamembrane, 26% cytoplasmic, and 3% nuclear; and *L*sppCaNB with 5% plasmamembrane, 49% cytoplasmic, and 16% nuclear. Interestingly, only the regulatory subunits *Tc*CaNB and *L*sppCaNB had 35% and 8%, respectively, presenting localization at the chloroplast level (Figure 8B).

### 3.5. CaN Regulatory Subunits of Obligate Intracellular Trypanosomatids Interact Only with Their Catalytic Monomer and Related Immunophilins

A PPI network was generated for each of the regulatory subunits; the red nodes represent the proteins of interest and the rest represent those with which it interacts (Figure 9); the score of each node represents the evidence of the interacting proteins. In order to validate the score obtained, a PPI network with *Hs*CaNB-1 was built. Interestingly, for *Tc*CaNB and *Lm*CaNB (and other orthologs in *Leishmania*, Appendix A), the substrate potential has not been determined, as well as for NFATC1 in *H. sapiens*.

### 3.6. CaN Regulatory and Catalytic Subunits of Obligate Intracellular Trypanosomatids Have a Differential Potential to Be Secreted by the Non-Classical Pathway

The predictions obtained for each of the subunits under study suggest that the subunits *Tc*CaNA1, *Tc*CaNB, and *L*sppCaNA2 with an NN-score of 0.682, 0.637, and 0.718, respectively, are secreted by the non-classical pathway. Only in the case of *Hs*CaNB-1 does it have an NN-score of 0.548, close to the threshold proposed for mammalian sequences (Table 4).

## 4. Discussion

Trypanosomatid parasites have evolved to survive in different environments as their biology has developed, either in vector insects or in mammalian hosts. These different environments involve different parasite forms that could have conditioned the appearance of specific trypanosomatid phosphatases [58]. Regarding CaN, through the analysis of the TriTryp phosphatome, two groups of CaN were identified; one of them grouped with *H. sapiens* and *S. cerevisae*, and the second with less similarity to eukaryotic CaN and with characteristic features of kinetoplastids, and presenting some mutations in its catalytic residues, suggesting that they could be pseudophosphatases [38]. CaN-like activities have been observed in vitro in some of these recombinant A2 subunits [33].

The domain structures of the catalytic subunits of *T. cruzi* (*Tc*CaNA1 and *Tc*CaNA2), in which each one of them presents the CaNB-BD and catalytic domain, have already been identified and characterized [32,33,37]. On the other hand, in *Leishmania*, the same analysis has determined that it possesses the four characteristic domains of CaN [37,59]. In the present work, we seek to highlight the differences at the level of the primary structure of the subunits under study and the proposed regulatory mechanisms. In this regard, mammalian catalytic isoforms have a molecular mass (MW) ranging from 57 to 59 kDa, whereas lower eukaryotes vary from 57 to 71 kDa [17]. In addition, the pI of the *Tc*CaNA2 and *L*sppCaNA2 subunits (8.13 and 6.37, respectively) deviate considerably from the range of pI observed for mammalian isoforms. In the case of the isoform α (pI = 5.6–5.8), β1 (pI = 5.3), β2 (pI = 5.6–5.8), however, the isoform γ (pI = 7.1) specific to the testis possesses a higher pI [60]. The latter suggests a differential subcellular distribution between the CaN subunits of these protozoa by virtue of pI [61,62].

A transcendent regulatory mechanism for the action of CaNA is cleavage by calpain. Curiously, calpains and calcineurins share the same subcellular distributions and exhibit equally high levels of activity after many of the same types of insults, thus Ca^2+^ and CaN-sensitive protease are considered common effectors of Ca^2+^-induced dysfunctions and degenerations [63]. Calpain is capable of cleaving CaNA directly and increasing its phosphatase activity [64], generating more active 57, 48, and 45 kDa products in excitotoxic neurodegeneration models [65]. From the potential calpain cleaved sites analyzed by GPS-CDD in the CaNA subunits of *T. cruzi* and *Leishmania* spp., only *Tc*CaNA1 and *L*sppCaNA1 possess a cleavage site located between the catalytic domain and the regulatory domain of CaN, which could upregulate CaN phosphatase activity by removing the regulatory domain [66] or parts of it [67], while the other sites could have other regulatory effects or impair CaN activity if they are located in the catalytic domain of CaNA. On the other hand, the insertion 243-VSGGSGSDYYTPSAGPSYGS-262 present in *Tc*CaNA2, but not in *Tc*CaNA1, has 4 of the 12 potential sites to be cleaved by calpain [33], while the insertion 235-YNNVEEPSGETYVPRLGLF-253 in *L*sppCaNA2 does not present any possibility of being cleaved. The regulation exerted by calpains in the biological cycle of *T. cruzi* has been studied through the use of inhibitors such as MDL28170, which stops the growth of epimastigote forms [68], in the same way as the role attributed to the *Tc*CaNA2 in epimastigote proliferation [33]. Moreover, using the inhibitor MDL28170, it was shown that it decreases the viability of blood trypomastigotes and affects metacyclogenesis as well as the adherence of the parasite to the luminal surface of the triatomine gut [69,70]. It also inhibits the growth and viability of promastigotes of *L. amazonensis* [71]. The involvement of these enzymes is so great that there is evidence to suggest trypanosomal calpains as good drug targets [72]. Despite this, it is important to consider that inhibitors such as MDL28170 also act on host calpains, which is why studies are required to determine how pseudoproteolytic trypanosomal calpains function and how calpain inhibitors act against them [73]. This reinforces the important role of molecules associated with Ca^2+^, such as calpains and CaN, in basic cellular functions of the life cycle of *T. cruzi* and *Leishmania* spp.

It has been described that CaNB-BD in the catalytic subunit αδ of rat brain has four hydrophobic amino acid residues (Val^349^-Phe^350^ and Phe^356^-Val^357^), which are essential for the interaction between the catalytic subunit and the calcineurin regulatory subunit [20]. This configuration allows two hydrophobic peaks to be generated in the CaNB-BD. In the case of CaNB-BD of the CaNA subunits of *T. cruzi* and *Leishmania*, the Val^349^ residue of the *Hs*CaNAα subunit is not present in their corresponding protozoan orthologs. This makes the hydrophobic profile of the surrounding residues different, which suggests that the degree of differential hydrophobicity of CaNB-BD present in the CaNA of *T. cruzi* and *Leishmania* conditions the interaction of the invariant CaNB subunit with the CaNA1 or CaNA2 subunits in the conformation of the heterodimers.

The diversity of CaNA in *Leishmania* that present or not the CaM-BD may establish a differential role in the cellular processes of the parasite. Interestingly, a calcium-independent, but CaM-CaN-dependent signaling pathway has been proposed in regulating the inversion of the flagellar wave, acting antagonistically to the cAMP-dependent pathway, so both of these pathways could establish an equilibrium between flagelar or ciliary churning waveforms, allowing appropriate motility and responses of the parasite in its environment, crucial for its viability, survival, and infectivity [35].

Unlike the CaM-BD present in *Hs*CaNAα, which has the following sequence: A**RK**EVI**R**N**K**I**R**AIG**K**MA**R**VFSVLREES, with conserved residues with positive charge (in bold) and potential kinase phosphorylation site (underlined) [21], which is recognized in the ARVFSVLRE context to be phosphorylated in S by an unknown kinase and in LREES by PKA and as well as in brain CaNA [74], the CaM-BD in *L*sppCaNA1 and *L*sppCaNA1_var have an identity of 50% and 53%, respectively, with only four residues with a positive charge; however, they have a greater potential to be targeted for phosphorylation by kinases such as PKA and PKC, either in serine or threonine. It may be a regulatory mechanism of the CaM interaction of the parasite towards the *L*sppCaNA1 and *L*sppCaNA1_var subunits, as has been described for other CaM-binding proteins [75,76].

In relation to the AID present in *Hs*CaNAα, it has been proposed as a pseudosubstrate exerting an action mechanism that blocks the active CaN site [24], together with evidence of the existence of additional autoinhibitory elements between CaM-BD and AID [77]. In fact, an autoinhibitory element present in the α and β isoforms of human CaNA called ASI (autoinhibitory segment) has been described, having the sequence ^416^ ARVFSVLR^423^; importantly, it interacts with a hydrophobic groove formed at the junction of the CaNA and CaNB subunits [78]. In LsppCaNA1 and LsppCaNA1_var, the amino acid sequence of this region was assigned as SRMFHTLC. This element could have a similar role in *Leishmania* spp., as the ASI potential presents three nonpolar residues (M, F, and L) of the total of five present in the ASI in the isoforms α and β of human CaNA, and the basic character of the R residue is present in both sequences.

In the case of CaNB, the domains’ architecture possessed by CaNB of these trypanosomatids is different, with three EF hand motifs for *T. cruzi* and four for *Leishmania* spp. [32,37]. These structural characteristics suggest a differential activation mechanism for their corresponding catalytic subunits, which could be closely related to the associated functional roles, such as invasion, proliferation, response to stress, and motility, among others [32,33,34,35]. Interestingly, the analysis of the domain architecture in *Trypanosoma rangeli*, a non-virulent trypanosome for mammals, suggests the presence of four EF-hand motifs with a potential role in the growth of epimastigotes, and of these four EF-hand motifs, only three fulfill the expected pattern of amino acid residues involved in coordinating Ca^2+^ [79]. In the case of *Hs*CaNB, the analysis using the Prosite program attributes a lower score to the EH-hand motives towards N-term, while these are higher towards C-term, according to what has already been evidenced [80]. Thus, CaNB in these protozoa, unlike its counterparts in mammals and fungi that have four complete EF-hand motifs, has only two complete typical EF-hands, presenting the following “motif signature”: Dx-[DNS]-{ILVFYW}-[DENSTG]-[DNQGHRK]-{GP}-[LIVMC]-[DENQSTAGC]-x (2)- [DE]-[LIVMFYW] (http://prosite.expasy.org/cgi-bin/prosite/ nicedoc.pl?PS00018 (accessed on 30 May 2020)), establishing a conformation pattern of one or two damaged or incomplete EF-hands in the case of *Tc*CaNB [32,37] and *L*sppCaNB, respectively, followed by two complete functionals. 

The conformation pattern of the EF hand motifs (the first damaged and odd followed by two functional ones) of *Tc*CaNB is also observed in oncomodulin and parvalbumin, proteins that are phylogenetically related to CaNB [81]. Therefore, these low affinity sites in the CaNB subunit of these protozoa would have a conformational role towards the CaNA subunits, as has been described for rat CaNB; high affinity sites are always saturated with Ca^2+^, while the low affinity sites are regulated by the concentration of Ca^2+^. Thus, the binding of Ca^2+^ to the low affinity sites affects the conformation of CaNB and, when associated with CaNA, induces a conformational change in the regulatory domain, which leads to the exposure of the binding domain with CaM. This conformational change, necessary for the partial activation of the enzyme in the absence of CaM, allows it to become fully active when associated with CaM [80,82].

Through the generation of mutants in the E (acid) residue at position 12 of each loop that binds Ca^2+^ to L (basic) of *Hs*CaNB, the individual role of each EF hand motif was determined (by eliminating the Ca^2+^ binding capacity in each of them), showing that those mutations in the EF-hand motifs 1 and 2 (towards N-term) suffered variations in their electrophoretic mobility depending on the presence or absence of Ca^2+^. Meanwhile, in those with mutations in EF-hand 3 and 4 motifs (towards C-term), no differences were observed in the electrophoretic pattern, suggesting that Ca^2+^ binding in EF-hand motifs 1 and 2 varies according to their concentration, dynamically modulating the enzymatic function, while EF-hand 3 and 4 would always be saturated with Ca^2+^, fulfilling a structural role [80]. Moreover, ^45^ Ca^2+^ exchange is faster in EF-hand 3 than EF-hand 4 mutated, indicating that these sites are not equivalent and that the effects on heterodimer formation are greater in the EF-hand 3; this could explain the differences observed in the two functional EF-hands of *Tc*CaNB, establishing a structural role for EF-hand 3 and a conformational role for EF-hand 4, and for *L*sppCaNB, a structural role for EF-hand 3 and 4 and a conformational role for EF-hand 1 and 2.

With respect to the subcellular distribution of CaN, in mammals, it associates predominantly in the cytoplasm and in the synaptosomal cytosol [83]. It has also been associated with synthetic vesicles, suggesting that CaN binds unilamellar vesicles, this being a Ca^2+^-dependent mechanism [84]. Regarding the NFAT activation pathway, it is shown that Ca^2+^ induces an association between CaN and NFAT, which results in the colocalization of both molecules in the nucleus [85]. It is important to mention the location and differential expression of isoforms of the CaNA, as is the case of neuronal isoforms α and β. Isoform α was visualized at the nuclear level, while isoform β was found in the cytoplasm of a wide variety of cells of the central nervous system (CNS) [86]. These results suggest that each isoform would present different sites of action in the neurons of the CNS and that this phenomenon has been conserved during evolution. A clear example of this differential functionality is that observed in mice lacking the isoform β, in which alterations are generated in the function and development on immune system [87], while the mice lacking the α isoform suffer kidney dysfunction [88]. Therefore, it is possible to affirm that CaN isoforms can be differentially expressed and/or the activity of each one of them can be differentially regulated, allowing their specific function.

Studies in yeast have shown that CaN dephosphorylates the transcription factor Crz1p in vitro and that its location is displaced from the cytosol towards the nucleus subsequent to the in vivo activation of CaN, demonstrating that CaN regulates the phosphorylation and localization of Crz1p, and identifies Crz1p as the first CaN substrate protein [30]. Furthermore, their observations reveal that the mechanism by which CaN regulates gene expression in yeast and mammalian cells is strikingly similar [30]. Curiously, in parasitic protists, they do not appear to possess yeast Crz1 or mammalian NFAT orthologs in their genomes and lack transcription factors [31], suggesting that the CaN cascade may function in ways other than factor-mediated regulatory mechanisms of transcription that control the virulence of parasitic protists owing to the considerable evolutionary distance compared with fungi [12]. However, some CaN targets have been proposed for *Plasmodium falciparum* including HSP90, actin-1, and phosphoglycerate kinase [89]; in *T. cruzi*, CaN would act during the process of cell invasion on protein targets of high molecular mass [32], while in *Leishmania* spp., it has not been addressed.

Regarding the characterization of some CaN subunits in kinetoplastids, this has been described in *Leishmania* spp. and in *T. cruzi* [58], and in *T. brucei*, there is information on the cellular localization by microscopy of the CaN subunits: for *Tb*CaNA1 (Tb927.9.1540) in cytoplasm and flagellar cytoplasm; for *Tb*CaNA2 (Tb927.10.6460) in cytoplasm, flagellar cytoplasm, and nuclear lumen; and for *Tb*CaNB (ID Tb927.10.370) in endocytic compartmen and cytoplasm (see TrypTag.org database). In the case of *Leishmania donovani*, CaN has a cytosolic localization in promastigote forms that depends exclusively on Ca^2+^ and CaM for its activity, which was biochemically evidenced [90]. In *T. cruzi*, the *Tc*CaNA1 subunit presents an evidently nuclear cellular localization in epimastigotes and amastigotes [37], and *Tc*CaNA2 presents a cytosolic localization in epimastigotes [33], and in the case of *Tc*CaNB, a predominant localization was observed to be cytosolic with some accumulation in the kinetoplast (data not shown). These analyses support a probable co-distribution of the *Tc*CaNA2 and *Tc*CaNB subunits.

Through the in silico analysis of the amino acid sequences of the CaN subunits in *T. cruzi* and *Leishmania* spp., in the CELLO2GO tools [52] and WoLF PSORT [51], we sought to predict subcellular localization. Oddly, all CaN catalytic subunits of *T. cruzi* and *Leishmania* spp. have a marked predisposition for plasma membrane over cytoplasmic localization and, in the case of CaNA2, a strong extracellular component. These characteristics suggest that these phosphatases may be secreted and behave as virulence factors acting synergistically or independently in infected cells [91,92,93]. In the case of the *Tc*CaNB, the analyses suggest that a localization in the chloroplast may be due to its proximity to the calcineurin B-like proteins present in plants, which have substitutions in their first EF-hand motif that seem to be unable to bind Ca^2+^ [94] with the second incomplete EF-hand motif and the third and fourth complete motifs analyzed in Prosite (Appendix A).

Studies have shown that several extracellular proteins can be exported without possessing a classical N-terminal signal peptide, such as FGF-1, FGF-2, IL-1, and galectins [53]. In the case of *T. cruzi* and *Leishmania*, studies indicate that they produce ectosomes, exosomes, and other soluble proteins not associated with vesicles from which they are released to deliver cargo on the host cells [95,96,97,98,99]. Studies on *Leishmania* spp. suggest that the exosome pathway is an unconventional protein secretion mechanism as most of the exosome proteins do not contain a predicted signal peptide [95]. The analyses carried out give a secretion potential through the non-classical pathway to *Tc*CaNA1, *Tc*CaNB, and *L*sppCaNA2 (NN-score > 0.6); in the case of *L*sppCaNA1 and *Hs*CaNB, they have a close score, which argues in favor of the potential of these molecules to modulate the host cell, as in the case of some phosphatases described in *T. cruzi* [100]; particularly, the *T. cruzi* secretome has revealed the presence of Ca^2+^ binding proteins such as calreticulin (gi|322823951) involved in host–parasite interaction, a calcium-binding protein (gi|487896), calmodulin (gi|10386) involved in cell signaling, and a calpain-type cysteine peptidase (gi|322830271) whose role is associated with proteolysis [97], with a characteristic of these proteins being the presence of EF-hand motifs. One of these proteins that has acquired a preponderant role as a modulator of the functions of the extracellular microenvironment is calreticulin (TcCRT), mediating the infectivity of the parasite to inhibit complement; it is also antiangiogenic and inhibits tumor development in vivo [101].

Several data have shown similar host responses against *Leishmania* and tumor progression [102,103]. Both parasite infection and tumor cells have to deal with the immune response to proliferate and survive, showing the importance of controlling immunological response favoring both models [104,105]. However, target-signaling mechanisms could function in opposite directions when activated by its heterologous molecules [106]. In this scenario, the release of an active CaN could participate by blocking or activating signaling pathways at far distances to promote resistance or susceptibility. This is not uncommon, as the discovery of the role of extracellular vesicles by transporting active molecules [107] shows a sophisticated and broad way of communication for parasites [9] and tumor cells [105]. Recently, *Hs*CaNB has acquired interest as it has been attributed anti-tumor roles, categorized as immunity-mediated killing (extracellular role) and direct pro-apoptotic killing (intracellular role) [108]. In their extracellular role, it was determined that *Hs*CaNB is a ligand of the αM integrin, which corresponds to a subunit of the heterodimeric integrin αMβ2, a receptor expressed mainly on the surface of innate immunity cells (macrophages, monocytes, neutrophils, and NK cells), suggesting that the levels of *Hs*CaNB in the serum are important for the maintenance of the innate immune response and the immune surveillance of cancer, through the activation of the monocyte–macrophage axis [109]. It was then shown that exogenous *Hs*CaNB uptake depends on the TLR4/MD2/CD14 receptor complex, indicating a second membrane receptor (TLR4) plays a critical role in innate immunity and its connection with adaptive immunity [110]. Recently, recombinant *Hs*CaNB (rhCNB) has been implicated in the inhibition of the proliferation of gastric cancer cells and hepatomas, through the induction of apoptosis and arrest of the cell cycle, as rhCNB promotes the expression of p53 and decreases the expression of cyclin B1 and cyclin-dependent kinase 1 (Cdk1), contributing to arrest in G2/M [111]. Through the generation of rhCNB truncates, the domain that mediates internalization called Trun3 was identified, which is captured by tumor cells and directed to tumors with almost the same efficiency as untruncated rhCNB, thus being a perfect antitumor candidate [112]. These studies suggest that these trypanosomatid CaNB subunits (particularly *Tc*CaNB owing to their potential secretion) could play a role as mediators of the host immune response (among other potential mechanisms of action yet to be elucidated) during parasite infection and tumor formation.

With regard to the protein–protein interactions determined for *Tc*CaNB and *Lm*CaNB based on the parameters used, the predictions establish contact between the CaNB subunits and the CaNA subunits, and between the CaNB subunits and peptidylpropyl isomerases. In particular, in the case of *Tc*CaNB, the predicted functional partners XP_810491.1 (allele of TcCLB.508413.40) and XP_808861.1 (allele of TcCLB.510755.138) are putative sequences that have a 98–99% identity with *Tc*CaNA1 [37], while STRING does not predict the interaction between *Tc*CaNB and *Tc*CaNA2, as evidenced in vitro by far-Wester blotting [33]. Nevertheless, the other predicted functional partners were proteins with a peptidylprolil isomerase function FKBP type, among them XP_810893 described as TcMIP, which is homologous to other FK506 binding proteins [113], behaving as a virulence factor that is secreted only by trypomastigotes [114], and those parasites exposed to the cyclophilin *Tc*CyP19–trialisin complex show a greater capacity to invade the host cell through activation the parasite CaN pathway [115]. Curiously, the localization of *Tc*CaNB is mainly in the vicinity of the flagellar pocket (unpublished results) in infective and replicative forms, strongly suggesting that these *T. cruzi* proteins are secreted by events restricted to the flagellar pocket [116] such as the case of cruzipain [117].

Finally, the search for a new potential target as a parasitic CaN should have the objective of developing drugs that act specifically on pathogens while being non-immunosuppressive [12]. Another important aspect is that the drugs act directly on CaN, to avoid the effects of CsA and FK-506 on the peptidyl-prolyl-cis-trans isomerase activity of their related immunophilins (as mediators of the interaction with CaN), which is why the structure-based design of a highly selective inhibitor directed to the catalytic site of CaNA or on CaNB [118,119] are strategies that could specifically target *T. cruzi* and *Leishmania.*

## 5. Conclusions

In this work, it was possible to establish that the domain structure is diverse among the catalytic subunits of CaN of intracellular trypanosomatids, thus establishing potential different post-translational regulation mechanisms observed in the analysis of cleavage by calpains, or by the phosphorylation patterns in the regulatory domains of the catalytic subunits, particularly in the A1 subunits in *Leishmania* (*L*sppCaNA1 and *L*sppCaNA1_var). In the case of regulatory subunits, the domain structure is different, with *L*sppCaNB being more similar to the human regulatory subunit (*Hs*CaNB-1) than to *Tc*CaNB, although the binding affinity for Ca^2+^ is conserved between *Tc*CaNB and *L*sppCaNB. On the other hand, although the coupling sites to the immunophilin–immunosuppressive drug complexes are present in *Tc*CaNB and *L*sppCaNB, the myristoylation potential is only found in *Hs*CaNB-1.

Regarding the analysis of the subcellular distribution, the catalytic subunits of *T. cruzi* and *Leishmania* spp. preferentially (more than 50%) present a localization in the plasma membrane (with the exception of *Tc*CaNA2a with only 27.2%), different from *Hs*CaNAα, which is predominantly cytoplasmic (51.7%). In the case of regulatory subunits, the distribution was more heterogeneous, being more associated with chloroplast and cytoplasmic for *L*sppCaNB (33.8% and 26.5%, respectively) and preferentially in cytoplasm (48.6%) for *Tc*CaNB, when comparing the nuclear/cytoplasmic localization of *Hs*CaNB-1 (53.8% and 38.6%, respectively).

Besides the catalytic subunits, the interaction with molecules with peptidylprolyl isomerase activity, which is typical of cyclophilins and FKBPs, is confirmed in the analysis of *Tc*CaNB and *L*sppCaNB potential interactors. Concerning the potential secretion, *Tc*CaNA1, *Tc*CaNB, and *L*sppCaNA2 can be secreted by the non-classical pathway, suggesting new extracellular roles for these protein phosphatases.

On the basis of these in silico data, differential CaN regulation mechanisms are established between these protozoa and its human counterpart, complementing the knowledge of this phosphatase, promoting the development of new potential pharmacological targets to combat neglected diseases caused by these intracellular trypanosomatids.

## Figures and Tables

**Figure 1 biomolecules-11-01322-f001:**
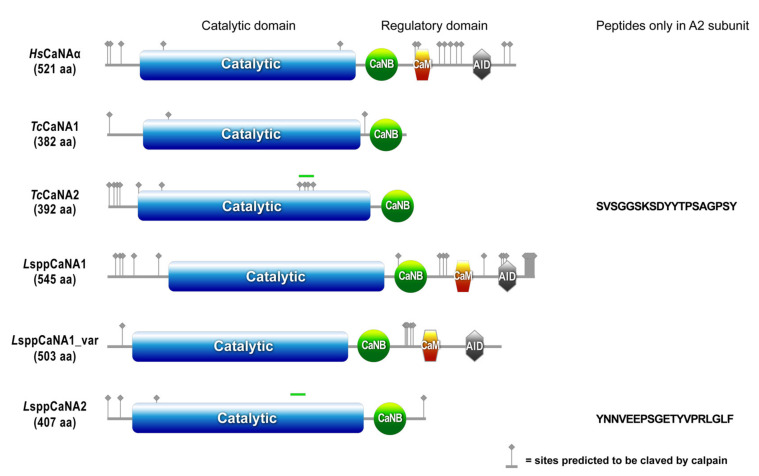
Protein domain architecture in CaNA. In blue is the catalytic phosphatase domain, in green is the CaNB-binding domain, in orange is the domain of interaction with CaM, and in gray is the AID. They are represented by the sites predicted to be cleaved by calpain by GPS-CDD, and the green band on the catalytic domains of *Tc*CaNA2 and *L*sppCaNA2 corresponds to a peptide not present in the CaNA1 subunits, specified to the right of each subunit. The image was prepared with the MyDomains image creator (Prosite).

**Figure 2 biomolecules-11-01322-f002:**
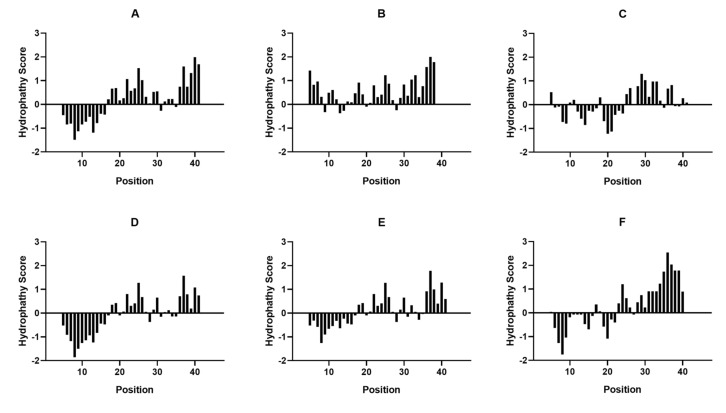
Hydropathy plot for CaNB-BD of CaNA preparated in the Expasy Protscale Website according to the Kyte and Doolittle algorithm. (**A**) CaNB-BD in *Hs*CaNAα, (**B**) CaNB-BD in *Tc*CaNA1, (**C**) CaNB-BD in *Tc*CaNA2, (**D**) CaNB-BD in *L*sppCaNA1, (**E**) CaNB-BD in *L*sppCaNA1_var, and (**F**) CaNB-BD in *L*sppCaNA2.

**Figure 3 biomolecules-11-01322-f003:**
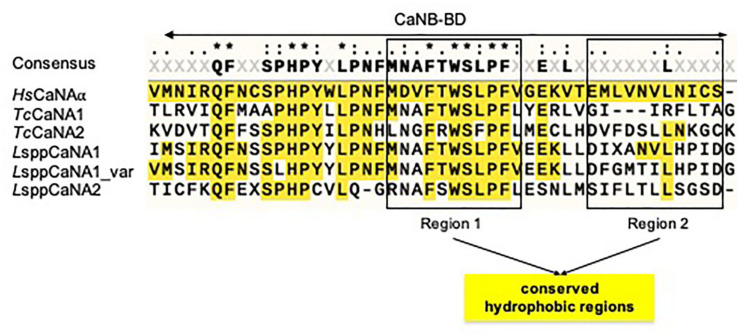
CaNB-BD sequence alignment present in CaNA. The boxes on the left and on the right correspond to the two hydrophobic regions present in CaNB-BD.

**Figure 4 biomolecules-11-01322-f004:**
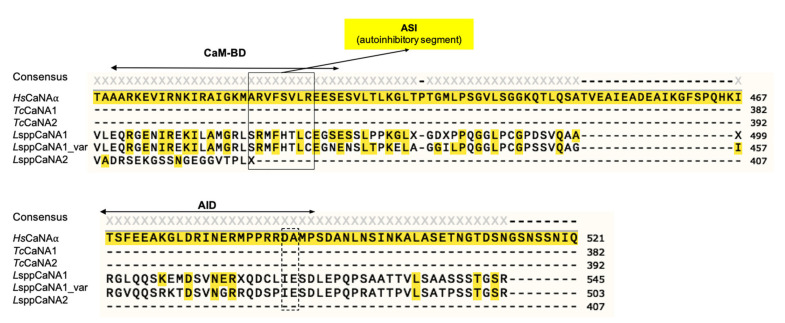
On top: CaM-BD sequence alignment present in CaNA. Underneath: AID sequence alignment present in CaNA.

**Figure 5 biomolecules-11-01322-f005:**
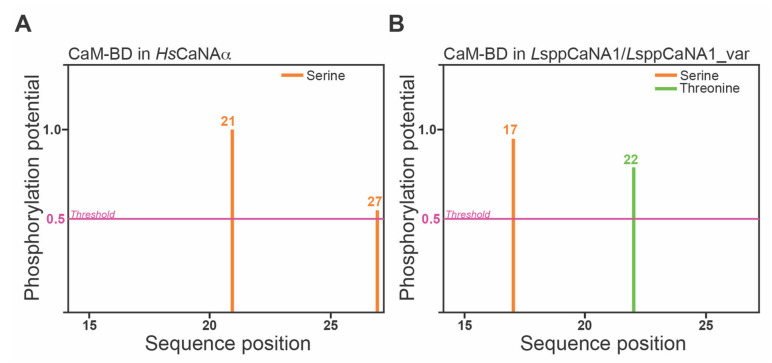
Phosphorylation sites of CaMB-BD in the catalytic subunits predicted by NetPhos 3.1 Server software. All phosphorylation sites presented at the different sequence positions (axis *x*) are represented as the phosphorylation potential score (PPS) over the threshold at 0.5 (pink line) at the axis *y*. The predicted phosphorylation sites for CaM-BD in *Hs*CaNAα (**A**) and in *L*sppCaNA1 and *L*sppCaNA1_var (**B**). Phosphorylation sites in serine (orange) or threonine (green) are shown.

**Figure 6 biomolecules-11-01322-f006:**
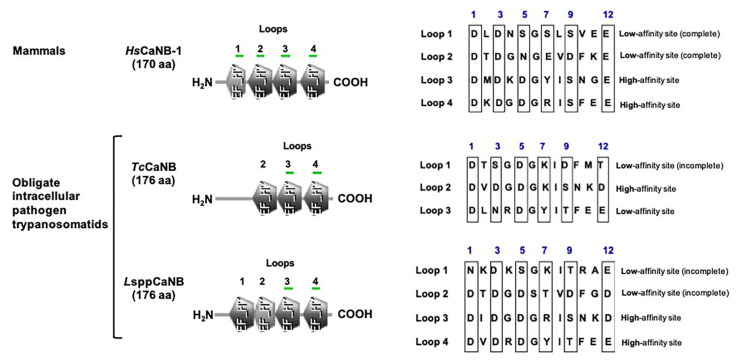
Protein domain architecture in CaNB. Representative scheme of the EF-hand motifs in mammalian CaNB (*Hs*CaNB-1) and in the obligate intracellular trypanosomatids (*Tc*CaNB and *L*sppCaNB) showing the loops (numbered) containing the calcium affinity binding sites (green underline) with the different amino acids sequences on the right. The blue number over the box represents coordinating sites for Ca^2+^.

**Figure 7 biomolecules-11-01322-f007:**
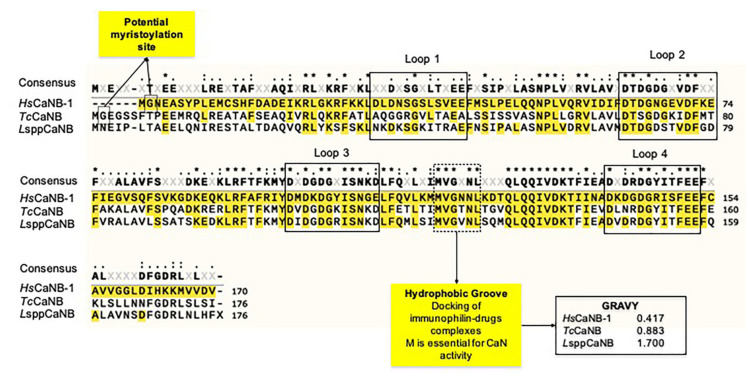
CaNB sequence alignment of the human CaNB (*Hs*CaNB-1) against the different obligate intracellular pathogen trypanosomatids (*Tc*CaNB and *L*sppCaNB). Conserved residues are in black with ***** (100% conservation), in black with**:** (>70% conservation), and in black with **·** (between 50 and 70% conservation). The consensus is shown with a threshold of 50%.

**Figure 8 biomolecules-11-01322-f008:**
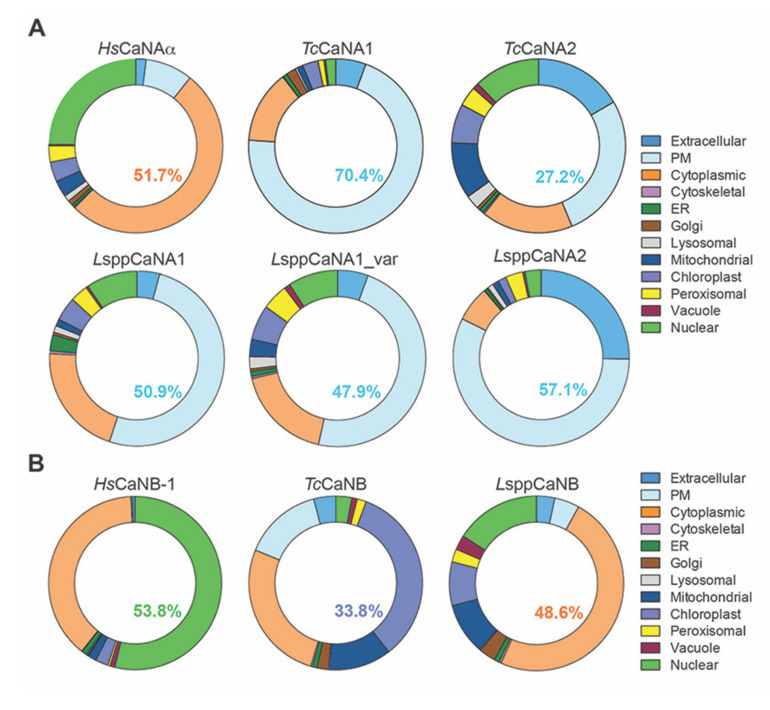
The predicted subcellular localization of CaNA (**A**) and CaNB (**B**) using the CELLO2GO web server. The subcellular localizations are represented in ring chart diagrams evaluating the significant terms in the form of their percentage contribution. In each one, the value with the highest probability is shown.

**Figure 9 biomolecules-11-01322-f009:**
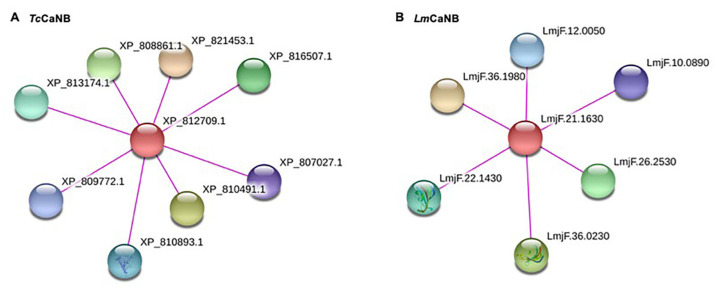
Protein–protein interaction network of the *Tc*CaNB (**A**) and *Lm*CaNB (**B**) proteins (STRING V.11.0). The line color indicates the type of interaction evidence (experimentally determined). Parameters: score (high confidence 0.7), sources of interaction used: experimental and databases. In A: XP_812709.1 (calcineurin B subunit, putative), XP_810491.1 (serine/threonine-protein phosphatase), XP_821453.1 (serine/threonine-protein phosphatase), XP_816507.1 (peptidylprolyl isomerase), XP_813174.1 (peptidylprolyl isomerase), XP_810893.1 (peptidylprolyl isomerase), XP_809772.1 (peptidylprolyl isomerase), and XP_807027.1 (peptidylprolyl isomerase); and in B: LmjF.36.1980 (serine/threonine protein phosphatase 2B catalytic subunit A2), LmjF.36.0230 (peptidyl-prolyl cis-trans isomerase, putative), LmjF.26.2530 (serine/threonine protein phosphatase, putative), LmjF.22.1430 (peptidylprolyl isomerase; Fk506-binding protein 1-like protein), LmjF.12.0050 (hypothetical protein, conserved), and LmjF.10.0890 (FKBP-type peptidyl-prolyl cis-trans isomerase).

**Table 1 biomolecules-11-01322-t001:** Overview of the physical and chemical parameters of CaN.

Protein	#aa	MW (Da)	pI
*Hs*CaNAα	521	58,687.85	5.58
*Tc*CaNA1	382	43,244.64	4.83
*Tc*CaNA2	392	44,619.50	8.13
*L*sppCaNA1	545	59,747.65	5.00
*L*sppCaNA1_var	503	55,559.36	5.61
*L*sppCaNA2	407	45,138.00	6.37
*Tc*CaNB	176	19,458.18	4.98
*L*sppCaNB	176	19,785.54	4.60

**Table 2 biomolecules-11-01322-t002:** Phosphorylation prediction results of CaM-BD in CaNA.

*Hs*CaNAα	#x	Context	Score	Kinase	Answer
# Sequence	21 S	ARVF**S**VLRE	0.974	unsp	YES
# Sequence	27 S	LREE**S**----	0.556	PKA	YES
** *L* ** **sppCaNA1 and *L*sppCaNA1var**	**#x**	**Context**	**Score**	**Kinase**	**Answer**
# Sequence	17 S	MGRL**S**RMFH	0.956	unsp	YES
# Sequence	17 S	MGRL**S**RMFH	0.725	PKA	YES
# Sequence	17 S	MGRL**S**RMFH	0.556	PKC	YES
# Sequence	22 T	RMFH**T**LCEG	0.766	Unsp	YES
# Sequence	22 T	RMFH**T**LCEG	0.517	CKII	YES

**Table 3 biomolecules-11-01322-t003:** Predicted subcellular localization of CaNs based on WoLF PSORT.

A. WoLF PSORT of Catalytic Subunits of CaN						
Site	*Hs*CaNAα	*Tc*CaNA1	*Tc*CaNA2	*L*sppCaNA1	*L*sppCaNA1_var	*L*sppCaNA2
extr						3
plas		1				
cyto	21.5	12	8.5	13.5	14.5	9.5
cysk				6		
E.R.	3	1				
golg		1				
mito		6	3	1	2	9
pero	3	2	5	3	1	2
vacu			1			
nucl	4	4	9.5	3	9.5	3.5
cyto_nucl	13.5	9	10.5	10	13.5	7
**B. WoLF PSORT of Regulatory Subunits of CaN**						
**Site**	** *Hs* ** **CaNB-1**		** *Tc* ** **CaNB**		** *L* ** **sppCaNB**	
extr	1				1	
plas			2			
cyto	25		5.5		9	
cysk	1		6			
mito			1			
pero	3		3		2	
nucl	2		9.5		15	
Cyto_nucl	14.5		8		14	

**Table 4 biomolecules-11-01322-t004:** SecretomeP 1. of predictions of CaNAs and CaNBs. Non-classically secreted proteins should obtain an NN-score/SecP score exceeding the threshold, but should not at the same time be predicted to contain a signal peptide.

Name	NN-Score ^1^	Odds	Weighted by Prior	Warning
*Hs*CaNAα	0.441	0.861	0.002	-
*Hs*CaNB-1	0.548	1.252	0.003	-
*Tc*CaNA1	0.682	1.863	0.004	-
*Tc*CaNA2	0.442	0.858	0.002	-
*Tc*CaNB	0.637	1.758	0.004	-
*L*sppCaNA1	0.528	1.094	0.002	-
*L*sppCaNA1_var	0.477	0.926	0.002	-
*L*sppCaNA2	0.718	2.165	0.004	-
*L*sppCaNB	0.382	0.709	0.001	-

^1^ The recommended thresholds are 0.5 for bacterial sequences and 0.6 for mammalian sequences.

## Data Availability

All the results used in this work to support the conclusions of this study are included in the article.

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
