# Peer review of "In Silico Characterization of Calcineurin from Pathogenic Obligate Intracellular Trypanosomatids: Potential New Biological Roles"

_biomolecules, 2021, doi:10.3390/biom11091322_

Round 1

Reviewer 1 Report

Orrego and cols. provided an in silico-based characterization of Calcineurin from T cruzi and Leishmania spp parasites, compared to the human version. They concluded that strutctural differences pointed to evidences of divergent regulatory mechanisms and activities among the forms tested. In addition, they predicted (I) non-canonical secretpry pathways of these calcineurins from parasites and (II) distinct subcellular distribution. The manuscript is well organized, although deserves an extensive English revision. The main weakness is the need for experimental evidences to support the conclusions made. The current version of the manuscript needs a careful and detailed review in order to be published. Please find below my point by point suggestions to authors.

  • Concerning the predicted subcellular distribution of CaN, please provide previously published evidences that corroborates the main findings.There is a very useful tool on TriTrypDB to identify genes based on Cellular Localization Imaging. Please go to www.tritrypdb.org > Protein targeting localization > Cellular localization Imaging. The assays were carried out in T. brucei, but it can be useful for inferences and experimental validations.
  • "CaN and CaB may have the potential to be secreted..." 

    Presenting a secretion signal is not a guarantee of communication function. Authors should provide more evidences in order to support the above mentioned statement

  • Please clarify the link between the main findings and the calpain/calcineurin as drug targets. 

  • Several conclusions are made based on the promastigote forms and little is discussed about amastigote-specific roles. Is there any differences in CaN activities based on your findings that could point for differences between pro and ama?
  • lines 429-431: be precise or remove the entire statement
  • Protein network analysis was carried out using Lmajor data, what about the orthologs from other species? Is there any insights that could enrich your analysis?

  • HsCaNAβ is mentioned in discussion, although, this isoform was out of your comparison studies
  • Line 248: “being much more hydrophobic in TcCaNB and LsppCaNB than in HsCaNB-1.” - How did you come to this conclusion?
  • Lines 263-265: “Interestingly, despite the fact that the domain structure between HsCnB-1 and LsppCnB is similar to each other, CaNB in Leishmania spp., lacks G in its N-terminal region to be myristoylated (Figure 7).” - On the basis of what result is this similarity? In my opinion it doesn't look so similar.
  • Lines 267-269: “HsCaNB-1 was predicted to be potentially myristoylated with a score of 0.98984294 (high confidence), while TcCaNB was predicted to be non-myristoylated with a score of -0.146144 (Table 3).” - The table shows that the HsCaNB-1 score is 0.00506406.
  • Lines 277-280: “while the CaNB subunits have a distribution potential predominantly in the cytosolic (cyto) and cyto_nucl compartments, having a marked distribution at the nuclear level (nucl) of CanB from T. cruzi and Leishmania (Table 4). - According to the second table 4, HsCaNB-1 shows cyto predominance and was not mentioned in the text. 
  • Lines 442 - 444: “establishing 442 a conformation pattern of one or two damaged or incomplete EF-hands in the case of TcCaNB [32,37] and LsppCaNB, respectively, followed by two complete functionals.” - I was unable to identify this in the figure 6.
  • Authors mentioned that CaNB is involved in inhibiting cancer cell proliferation. There are several evidences for similarities in Leishmania and cancer. In terms of mode of action, could the authors put in context the comparison between the two scenarios?

Minor

English is poor

Page 3 - Please update the T cruzi gene IDs as you did for Leishmania major. Also put all scientific names and genes in italic throughout the text

“was used” repeated in lines 152-153.

Figure 1 legend: "the orange band on the 182 catalytic domains of TcCaNA2 and LsppCaNA2"... Green?

Info presented in Table 2 can be included in Figure 5

Table 3 info can be embedded in the text. No need for a Table. The same logic applies to some figures. Authors must reconsider including some in supplementary material

There is no need to depict in details all the percentages on predicted subcellular localization within the text. Authors could only focus on the main results.

Color legend is duplicated in Fig 8. Remove one of them

There are too many repeated results in discussion section. Please rewrite it removing the redundant information.

Lines 415-418. There are 5x "of" in the same phrase. Too much "interestingly" even if the information is not that important or for negative results

Discussion is extremely (unnecessary) long! Authors must reduce it.

The human CaN must be included in Table 1

Line 222: “In total, 2 serines (position 21 and 27) have a potential to be phosphorylated in the CaM-BD of HsCaNAα (Figure 5A)”. - Position 27 is not shown on Figure 5A

Line 241: “HsCaNB-1 and LsppCaNB have two low loops and two high affinity loops for Ca2+, while TcCaNB only has one of each type (Figure 6).” - The image says that TcCaNB has 2 high affinity and not 1.

Tables 4: merge them or indicate they are different tables

Reviewer 2 Report

In this study Orrego et al. performed an in silico characterization of calcineurin a calcium dependent serine/threonine protein phosphatase in Trypanosoma cruzi and Leishmania, that acts as a crucial connection between calcium signaling and the phosphorylation states of numerous important substrates.

This manuscript has an objective important contribute to understanding the differential calcineurin regulatory mechanisms, subcellular distribution, interactors, and catalytic and regulatory subunits of the parasites, and comparing them to human calcineurin. These analyses and data are important to promote the development of new potential pharmacological targets against the chagas disease and leishmaniasis, two neglected diseases. The paper is well written and can be recommended for publication after minor revision.

Comments

In the abstract the parasite species should be related to the results.

Methodology

Which Leishmania species were studied to obtain the amino acid sequences? Please specify the Leishmania species in the items of the methodology

Which parasite form was used to obtain the consensus amino acid sequences of the catalytic regulatory sub-units of T. cruzi and Leishmania? (Line 107).

Which parasite form was used to identify interactions of the enzyme subunits of T. cruzi and Leishmania? (Line 127).

Results

Please rewrite the description of myristalization results (Fig. 7).

Please explain: “The analysis of all the sequences of the catalytic subunits under study using WoLF SPORT suggests a distribution predominantly in the cytosolic, nuclear and cytosolic-nuclear compartments” (lines 274-275). Predominantly?

Discussion

The discussion of the enzyme isoforms in the Purkinje cells of the cerebellum should be reduced (line 482).

In the case of L. donovani, the enzyme location was determined, this being the one with a cytosolic location in amastigotes or promastigotes? Please explain.

The authors should consider reducing the discussion related to several extracellular proteins exported without possessing a classical N-terminal signal peptide (line 526).

Despite being a drug target, the immunosuppressive nature of the current calcineurin inhibitors is an obstacle to overcome. Which experimental and theoretical strategies can be useful in development of novel drugs that can specifically target T. cruzi and Leishmania? Please discuss.

Reviewer 3 Report

This is an in silico analysis of Calcineurin subunits from Trypanosoma cruzi and Leishmania spp. By comparing them to the human ortholog, several predictions were made regarding their post-translational regulatory mechanisms, subcellular distribution, interactors, and ability to be secreted. While this study is of value to those who are interested in calcium signaling in trypanosomatids, the lack of experimental evidence makes it ultimately unsuitable for publication in this journal.

Author Response

We regret the reviewer's decision, but our job is what it is. An in silico comparative analysis manuscript of the CaN of intracellular parasitic trypanosomatids (T. cruzi and Leishmania spp.) With human CaN, supported by previous experimental investigations by us and other researchers. All this thinking that CaN of these parasites could become a potential chemotherapeutic target.